# Health Implications of Virtual Architecture: An Interdisciplinary Exploration of the Transferability of Findings from Neuroarchitecture

**DOI:** 10.3390/ijerph20032735

**Published:** 2023-02-03

**Authors:** Cleo Valentine

**Affiliations:** Department of Architecture, University of Cambridge, Cambridge CB2 1PX, UK; crv29@cam.ac.uk

**Keywords:** virtual reality, VR, virtual architecture, neuroarchitecture, environmental health

## Abstract

Virtual architecture has been increasingly relied on to evaluate the health impacts of physical architecture. In this health research, exposure to virtual architecture has been used as a proxy for exposure to physical architecture. Despite the growing body of research on the health implications of physical architecture, there is a paucity of research examining the long-term health impacts of prolonged exposure to virtual architecture. In response, this paper considers: what can proxy studies, which use virtual architecture to assess the physiological response to physical architecture, tell us about the impact of extended exposure to virtual architecture on human health? The paper goes on to suggest that the applicability of these findings to virtual architecture may be limited by certain confounding variables when virtual architecture is experienced for a prolonged period of time. This paper explores the potential impact of two of these confounding variables: multisensory integration and gravitational perception. This paper advises that these confounding variables are unique to extended virtual architecture exposure and may not be captured by proxy studies that aim to capture the impact of physical architecture on human health through acute exposure to virtual architecture. While proxy studies may be suitable for measuring some aspects of the impact of both physical and virtual architecture on human health, this paper argues that they may be insufficient to fully capture the unintended consequences of extended exposure to virtual architecture on human health. Therefore, in the face of the increasing use of virtual architectural environments, the author calls for the establishment of a subfield of neuroarchitectural health research that empirically examines the physiological impacts of extended exposure to virtual architecture in its own right.

## 1. Introduction

The last five years have seen the rapid proliferation of virtual architecture [1], facilitated, in large measure, by advances in virtual reality technology, and catalysed by the nascent ’metaverse’ [2]. The metaverse is a “future iteration of the internet, made up of persistent, shared, [three-dimensional] 3D virtual spaces [e.g., virtual architecture] linked into a perceived virtual universe” [3] (p. 1). While still in its infancy, the metaverse has attracted considerable attention, with companies such as Meta (formerly Facebook) and Microsoft investing over $80 billion USD into its development in 2021 [4]. There have been similar investments made by major architecture firms, including Zaha Hadid Architects’ cyber-urban ‘Liberland Metaverse City’ [5] and Bjarke Ingels Group’s (BIG) virtual office design ‘Viceverse’ [6]. Children and young adults are increasingly exposed to virtual architecture through the growing use of VR in educational [7] and recreational settings [8].

Virtual architecture (VA) refers to three-dimensional, computer-generated architectural spaces [i.e., buildings] which can be explored and interacted with by a user only when using fully immersive virtual reality devices [i.e., virtual reality goggles] [9] (For the remainder of this paper, VA does not refer to the interactions between individuals within these spaces, but the interactions of individuals with these spaces and the architectural forms that define these spaces. As a result, the vast body of scholarship examining the physiological responses to video games [8,10] or 3D virtual worlds such as Second Life or Fortnite [3,11] does not pertain to this paper.). To date, VA has been primarily used by architects to facilitate tasks such as structural analysis training [12,13] and user behaviour analysis [14,15]. However, VA has also been heavily relied on to evaluate the health impacts of physical architecture (PA) [i.e., architectural designs which are both constructed using tangible materials and physically inhabitable] [16,17,18,19,20]. These health impacts include both physiological and psychological impacts, ranging from changes in mood and stress levels to elevated levels of neuroinflammatory activity [21,22,23]. There is some emerging evidence that PA may even increase the risk of developing neurodegenerative disorders [23]. In this health research, exposure to VA has been used as a proxy for exposure to PA. Throughout the remainder of this paper, these studies are referred to as ‘proxy studies’. While proxy studies indicate that PA may elicit beneficial health outcomes [24,25,26,27], they also indicate associated risks [28,29,30]. As these studies use VA as a proxy, their findings may also predict the potential positive and negative health impacts of exposure to VA over short periods. Therefore, these studies are a useful starting point for understanding the health risks associated with VA.

However, there may be additional confounding variables that occur during long-term exposure to VA that are not captured by proxy studies. On this point, there has been a notable lack of independent research. We can only speculate as to the reason; it may be that it is assumed that proxy studies capture the impact of VA on human health or simply that the growth of virtual environments is too recent to have attracted academic inquiry. Additionally, the field of neuroarchitecture is in its infancy and is growing in response to developments in the fields of both neuroscience and architecture. In light of the paucity of scholarship examining the health impacts of extended exposure to VA, this paper considers: what can proxy studies, which use VA to assess the physiological response to PA, tell us about the impact of extended exposure to VA on human health?

This paper argues that the applicability of these findings to VA may be limited by certain confounding variables when VA is experienced over extended periods. In doing so, the ground is set for future empirical testing of these hypotheses (explored further in Section 7). More specifically, this paper identifies two potentially confounding variables, multisensory integration and gravitational perception. These confounding variables appear to be unique to extended VA exposure and may not be captured by proxy studies. Consequently, we should not assume that architectural forms are experienced identically in both the physical and virtual worlds.

The primary aims of this paper are twofold: (i) to assess the degree of generalisability between VA and PA and (ii) to identify some of the confounding variables that may limit the transferability of findings from proxy studies to VA. This analysis demonstrates the need for independent research into the health implications of VA, and provides a theoretical foundation for future empirical research.

An interdisciplinary and exploratory method is engaged to answer the research question and pursue the above-stated aims. A desk review of the literature from the adjacent fields of sensory cognition and gravitational continuity is synthesised with existing literature on the health impacts of exposure to PA from the field of neuroarchitecture. This interdisciplinary study produces a more encompassing understanding of the impact of VA on human health, providing a hypothesis for empirical testing. It is hypothesised that the two identified confounding variables—multisensory integration and gravitational perception—may produce negative health impacts during long-term exposure to VA that are not captured by proxy studies.

In conclusion, this paper argues that while proxy studies may be suitable for measuring some aspects of the impact of both PA and VA on human health, they may be insufficient to fully capture the unintended consequences of extended exposure to VA on human health. Therefore, in the face of the developing metaverse and the increased uses of virtual architecture in everyday life, the author calls for the establishment of a subfield of neuroarchitectural health research that empirically examines the physiological impacts of extended exposure to VA in its own right. Lastly, it encourages designers of VA to be cognisant of the impact of architectural forms on human health and the confounding variables that influence how we experience VA.

## 2. Use of Virtual Technologies by Children, Adolescents or Young Adults

The health implications of VA are, arguably, of particular relevance to children, adolescents and young adults. This demographic is increasingly exposed to virtual environments, particularly through gaming [8] and schooling [7]. Through these environments, children, adolescents, and young adults will inevitably come into contact with VA. Recent research from the Institution of Engineering and Technology (IET) [31] reports that more than a fifth of 5–10-year-olds (21%) own a VR headset, 15% have already tried VR, and 6% use it on a regular basis. Moreover, the IET predicts that the next generation will spend approximately ten years in virtual reality in their lifetime, equating to roughly 2 h and 45 min per day [31].

In response, scholars have begun to study the ethics, health, and safety of VR use among children and adolescents from an interdisciplinary perspective [32]. A systematic review by Kaimara et al., 2021 highlights that this research has been divided. For example, Kaimara et al., 2021 show that a number of studies have highlighted the potentially positive impacts of virtual reality use on cognitive, motivational, emotional, social development, and learning [33,34,35,36,37,38,39,40,41,42,43,44,45,46,47]. Conversely, there is evidence that VR has the potential to elicit strong negative emotional consequences in adolescents [8], disrupt circadian rhythms, and increase the likelihood of addiction, cardio-metabolic deficiencies and obesity [48,49,50,51,52,53,54]. Further research has provided evidence to suggest that the use of VR in adolescents can result in cybersickness and confusion between real and virtual worlds [55,56,57,58,59,60]. Yet, no systematic investigation has been conducted into the impact of VA on the physiological and psychological well-being of adults, adolescents, or children. Given adolescents and children’s increased exposure to VR and the exponential growth of VA more generally (discussed in Section 3 below), this study attempts to set the groundwork for future academic inquiry into the impact of VA on human health.

## 3. The Uses of Virtual Architecture

Over the past two decades, VA has been increasingly engaged by the architectural, engineering, and construction (AEC) industry to provide low-cost 3D renderings of buildings prior to construction [61]. This has allowed professionals to visualise and experience buildings during the design process and easily communicate complex ideas to clients [61]. Advancements in computer graphics and hardware have also widened the functional applications of VA in research and industry with additional uses including architectural education [62,63,64], construction safety and training [65,66], structural analysis training [12,13], emergency training and rescue [67,68], and user behaviour analysis [14,15].

More recently, virtual architecture has also been used by architectural health researchers to understand the neurophysiological effects of spatial design [26,28,69]. This research has been facilitated by advancements in neurotechnology equipment [i.e., devices which allow for the brain to be interfaced with electronic devices [70]], which have increased both the functional and financial accessibility of tools such as electroencephalography (EEG) devices, which allow for researchers to evaluate changes in electrical activity within the brain.

Outside of the AEC industry, VA has become part of mainstream platforms and services. For example, museums, including the Victoria and Albert Museum in London, the Louvre in Paris, and the Smithsonian in Washington, DC, have featured VA in exhibitions [71]. Perhaps the most topical platform for VA, however, is the ‘metaverse’. The metaverse is a “massively scaled interoperable network of real-time rendered 3D virtual worlds [i.e., virtual built environments] which can be experienced synchronously…by an effectively unlimited number of users”’ [2]. Unlike the past iterations of the internet, the metaverse is inherently ‘spatial’ in that it involves placing users inside a ‘virtual’ or ‘3D’ version of the internet [72,73,74]. Most notable is the release of Meta’s ‘Horizon’ environments, which include virtual homes, office spaces and meeting rooms, and cityscapes which allow users to design and inhabit virtual environments. Since its release in early December 2021, Horizon’s monthly user base has increased tenfold and now includes 300,000 users [75].

The metaverse’s spatial nature and dependency on virtual environments has arguably catalysed the importance of VA. Though the metaverse is still in the early stages of development (it is not expected to be fully operable for another 5–10 years [74]), it has already significantly impacted the architectural profession. In June 2021, the world’s first virtual house, designed by Krista Kim, was sold for over $500,000 USD [76]. The development of ‘digital land’ has further increased the value of VA by providing real estate markets for these digital buildings. Digital real estate sales in the metaverse totalled over $500 million USD in 2021 and are predicted to double in 2022 [77]. According to the Crypto asset management firm Grayscale, virtual real estate has the potential to generate $1 trillion USD in annual revenue in the future [78]. According to Gabe Sierra, the founder of the digital real estate development group ‘Meta Residence’, “as experiences become more and more immersive, the lines will continue to blend until reality and the metaverse is nearly indistinguishable” [79].

Despite the fast-paced development and adoption of VA, there is a notable gap in the literature examining the physiological responses to these spaces. Consequently, very little is known about the potential health implications of extended exposure to VA. This paper aims to set the foundation for further empirical research on the health impact of VA by providing hypotheses for future testing.

## 4. Virtual Architecture as a Proxy for Measuring the Health Impacts of Physical Architecture

Although the field of neuroarchitecture has yet to examine the direct effects of prolonged exposure to VA on human health, neuroarchitectural researchers have frequently utilised VA to evaluate the health effects of visual exposure to PA. These studies may provide some insight into the positive and negative impacts of VA on both physiological and psychological wellbeing (referred to here collectively as health impacts). These studies may provide some insight into the positive and negative impacts of VA on both physiological and psychological wellbeing (referred to here collectively as health impacts). However, this paper proceeds to consider evidence from the adjacent fields of sensory cognition and gravitational continuity to propose that exposure to VA may produce unique impacts on human health. Consequently, it hypothesises that prolonged exposure to VA may produce health impacts beyond those captured by existing studies examining the impact of PA on human health despite these studies having used VA to simulate the effects of PA. Before advancing to consider the literature on sensory cognition and gravitational continuity in relation to virtual environments (Section 5), it is useful to review existing findings from architectural health research. This research, outlined below, has been facilitated by technological advances in virtual reality equipment and clinical biosensors (clinical biometric sensors refer to medical devices that collect objective empirical measurements of bodily functions which predict clinically relevant processes [80].). These advancements have allowed researchers to use VA as a tool to isolate and readily adjust specific VA features (i.e., window placement, lighting levels and spatial composition) while measuring performance factors and physiological responses safely and effectively [81]. In these instances, visual exposure to VA has been used as a proxy for measuring various physiological responses following visual exposure to PA. The correlation between VA and PA has been borne out in the literature [16,19,81,82,83] and will be discussed in the coming section.

Scholarship in architectural health has often focused on examining more tangible environmental variables such as noise pollution [84,85], air quality and pollution [86,87], ventilation [88], water quality [89,90], material off-gassing [88,91], and ambient temperature [92,93]. More recently, proxy studies have examined the psychological and physiological responses to visible variations in architectural forms, such as window placement and wall curvature [24,28,94]. Both psychological and physiological responses contribute meaningfully to human health [95]. This paper focuses on the impact of architecture on physiological stress responses as they have been found to contribute to the development of psychological disorders, such as depression, anxiety, and schizophrenia [96], and have been the subject of less academic inquiry. Physiological stress responses are defined as a “threat, real or implied, to the psychological or physiological integrity of an individual” [97] (p. 108).

These proxy studies have provided compelling evidence of a connection between isolated architectural features and physiological stress responses [18,21,24,26,28,69,94,98]. For example, studies examining the stress-reducing effects of architecture have illustrated the health benefits of visual exposure to features including biophilic architectural design (biophilic design refers to spatial designs which integrate live organic material [99].) [22,26,100], biomorphic architectural design (biomorphic design refers to designs which “imitate the contours or motifs of organisms” of organic matter [99] (p. 16).) [101,102,103], window size and placement [22,98], daylight [104], navigation signage [24,105], rectilinear and curvilinear architectural form [106], and viewing location [98]. Conversely, a growing body of research outlining the stress-inducing effects of architectural features has evidenced that exposure to varied room proportions [28,29], wall curvature [29,30], lighting conditions [24], and window arrangement and size [28,94], regularly provoke stress responses in humans without their conscious perception.

A review of the literature reveals just a single study that directly (referred to herein as ‘direct study’) examines the potential health implications of visual exposure to VA. In this study, Ashley Verzwyvelt et al. (2021) examined if exposure to virtual biophilic architectural design could decrease oncology patients’ pain and distress while receiving chemotherapy. In the study, 33 cancer patients were randomly placed in one of three different rooms to receive chemotherapy: control room, physical biophilic room, and virtual biophilic room. The study found that patient heart rate, blood pressure, and self-reported distress levels decreased after exposure to the physical and virtual biophilic designs; however, these findings were statistically insignificant. The transferability of this study’s findings is further limited as the study involved cancer patients, not healthy individuals, and involved only an acute duration of exposure. However, this study emphasises the need for future research to carefully balance the negative effects of VA against the positive effects in order to appreciate the full health consequences associated with prolonged exposure to VA. It is possible that the positive effects of VA outweigh the negative effects of confounding variables, which are considered later, depending on the circumstances in which VA is being employed.

On account of the paucity of direct studies, this paper considers what proxy studies may tell us about the impact of VA on human health. The author considers that proxy studies, although informative, may be of limited transferability due to certain confounding variables that result from the unique features of VA use and design (considered in Section 5). As highlighted in Figure 1 below, two of these confounding variables, multisensory integration and gravitational perception, are identified and analysed in further detail. However, it is important to note that additional confounding variables not explored here may exist, highlighting an opportunity for future research.

## 5. The Generalisability of Proxy Studies

The reliability and efficacy of proxy studies depend on establishing a high degree of generalisability between physiological responses to VA and PA [18,81,107]. Generalisability can be defined as “the extension of research findings and conclusions from a study conducted on a sample population to the population at large” [108] (p. 1). In the context of this paper, generalisability refers to the degree to which human responses to VA replicate human responses to PA. A high degree of generalisability would suggest that subjects respond similarly to identical architectural features in virtual and physical settings. Establishing a high degree of generalisability is critical, as it determines the degree to which VA can act as a proxy for PA. In response, a handful of studies have emerged evaluating the consistency of physiological response data across identical PA and VA settings [16,19,81,82,83,109]. These studies have found that VA is a reliable proxy for PA. For instance, using clinical biometric sensors, Kalantari et al. (2021) evidenced a high degree of consistency in cognitive performance and physiological responses between visual exposure to PA and VA, which suggests VA is a reliable and valid proxy for PA in studies examining acute neurophysiological responses to visual features in PA.

However, a smaller body of scholarship questions these findings. From this literature, it is possible to extract five primary limitations. First, to date, proxy studies have largely focused on examining visual responses to architectural forms. As a result, the more complex sensory interactions involved in spatial experiences have yet to be adequately explored [110]. Second, there is an established body of scholarship that has explored the shortcomings of VA in creating convincing user experiences. Notable concerns include perceived level of realism and experience dimensions [111,112], user-related factors [113], and navigational issues [83,114]. Third, there are several neurophysiological health concerns regarding the ‘equipment’ (i.e., virtual reality hardware) used in VA. Most notable are concerns regarding optical safety [45,115], and ‘re-entry’ or transition syndromes [116]. Fourth, architectural health is an intersectional domain with numerous interacting variables including, but not limited to, thermal comfort [93,117], access to natural light [118,119], noise pollution [120,121], acoustics [84,85], and material off-gassing and air pollution [122]. Lastly, to date, proxy studies have only assessed physiological stress responses during acute, short-term exposure to architectural forms in VA. Interestingly, none of the studies referenced above have measured physiological stress responses to chronic or extended exposure to VA, nor have they considered the effects of repeated exposure. This may be due to logistical limitations regarding participant involvement. The acute nature of these studies limits the relevance of their findings when considering extended exposure to VA.

## 6. Confounding Variables in Virtual Architecture

Despite the aforementioned shortcomings, proxy studies provide many valuable insights into PA’s positive and negative impacts on human health. The question for this paper is whether the findings of these proxy studies capture the full impact of extended exposure to VA on human health. While proxy studies provide valuable insights into the potential health risks and benefits of exposure to VA, the transferability of these studies’ findings to VA when experienced over extended periods may be subject to some limitations. Suggesting a lack of transferability, defined here as the ability to “transfer original findings to another context, or individuals” [123] (p. 1212), may appear unusual, given that VA has been established as a reliable and valid proxy for PA. However, an interdisciplinary review of scholarship from the adjacent fields of sensory cognition and gravitational continuity identifies confounding variables unique to VA. These variables, in turn, may mitigate or exacerbate the negative or positive health impacts of architectural forms when experienced in virtual environments. Consequently, this paper argues that transferring the findings from proxy studies to the virtual world may prove problematic as proxy studies may be incapable of capturing the full physiological impact of VA due to confounding variables. The following section will examine two of these confounding variables: multisensory integration and gravitational perception.

### 6.1. Multisensory Integration and the Unisensory Nature of Virtual Architecture

PA is multisensory [85]. The occupant of physical spaces receives visual, auditory, somatosensory, vestibular, and olfactory sensory inputs [124]. In PA, all sensory inputs are stimulated simultaneously [125]. For example, when listening to someone’s footsteps on wooden floorboards, estimates about their location can be derived from three distinct sensory network inputs: visual, auditory, and proprioceptive [126]. This synergy, or interaction, between sensory inputs and the combination of their information content, is known as multisensory integration [127]. As mentioned above, the ability of proxy studies to capture multisensory integration is limited. This limitation may present a unique problem for VA, as the interaction between VA, which is unisensory, and the PA inhabited by the user, which is multisensory, may disorient the user and produce negative physiological responses.

The visual-centric nature of VA may result from the extraordinarily complex programming and data hosting required to integrate multisensory qualities such as sound reverberating against surfaces, the texture of materials, thermal comfort, or smell [9]. However, this limited sensory input may result in the user being deprived of complementing sensory experiences (‘sensory deprivation’), as the optical neural network is activated while the auditory and proprioceptive sensory networks are not. Without secondary and tertiary sensory signals to corroborate and calibrate the visual signal, the user’s sensory integration is conflicted [128]. As a result, the complex physiological processes required to perceive spatial information may not work optimally in virtual environments [129]. Sensory Conflict Theory proposes that this sensory deprivation (i.e., lack of auditory and proprioceptive signals), results in physiological stress and the associated disorientation and discomfort [130]. These findings were corroborated recently by Marucci et al. (2021), who found that the more sensory inputs a user had, the lower their physiological stress levels were and the faster and more effective their ability to process and detect stimuli.

Similarly, discrepancies between inputs (“sensory discrepancy”) coming from VA and PA may also produce stress responses. Multisensory integration is limited by aberrations in sensory interactions between the user’s virtual and physical environment [129,131,132,133]. These discrepant sensory interactions may produce a sense of discomfort in the user [129]. Research on virtual reality has suggested that some users may subconsciously detect the sensory differences between the virtual 3D scene presented to them and the real world, causing them a profound sense of disorientation [129]. For instance, when a user receives visual input from a VA environment while receiving notable auditory or proprioceptive inputs from a PA environment, they may become aware of a sensory divergence between the two environments they are simultaneously inhabiting.

An example of this may be being ‘visually’ present in a VA environment that looks tranquil while physically being present in a PA environment with excessive noise pollution. In these instances, the Neural Mismatch Model proposes that when the combination of sensory inputs differs from the user’s past experience or expectation, a ‘mismatch’ can occur, which results in an inability to cognitively comprehend their environment [134]. This, in turn, may result in physiological stress and subsequent distress and discomfort [134]. These symptoms are often referred to as “cybersickness” or “simulator sickness” [135,136,137].

### 6.2. Gravitational Perception in Virtual Architecture

The transferability of proxy studies may be further complicated by emerging evidence of the intersensory nature of gravitational perception. Gravity provides the schema that dictates how matter behaves and what spatial forms can exist on Earth [138]. This schema has significantly influenced the evolutionary development of human neurophysiology, as all humans evolved under a gravitational constant of approximately 9.81 m/s^2^ [139]. As a result, humans have developed a combination of sensory systems which allow us to perceive spatial gravitational cues and respond accordingly [139]. This sensory system relies on both visual and bodily, such as tactile and vestibular, cues to predict changes in gravitational forces [138].

Emerging research suggests that visual perception plays an especially critical role in gravitational perception [72,140]. This research has evidenced that visual perception is not simply based on incoming visual signals, but also on multisensory vestibular-proprioceptive (i.e., the perception of the body’s movement through space) input and motor signals [72]. The result of this intersensory dependency is that even subtle deviations in visual cues of gravity (‘visual gravity’) may result in sensory conflict, and subsequently, may elicit negative physiological stress responses [141]. For example, Cano Porras et al. (2020) found that downhill visual cues elicit downregulatory anticipation responses (i.e., predicting the need to break in anticipation of gravitational acceleration going downhill). In contrast, uphill visual cues elicit upregulatory exertion responses (i.e., predicting the need to increase exertion in anticipation of gravitational deceleration going uphill). These findings are significant, as they illustrate how visual perception of gravitational forces can elicit upregulatory or downregulatory physiological stress responses. It can be argued that, when applied to VA, these findings suggest that exposure to buildings with slight disruptions in the clarity, consistency, and predictability of visual gravity may result in physiological adaptation. However, unlike PA, VA is not bound by Earth’s gravitational force. In VA, designers can create buildings that are not constrained by the same laws of gravity. For example, while the geometric composition of a bridge in the physical world is bound by the Earth’s thermodynamic forces, the same bridge built in virtual reality may defy gravity. In these instances, visual sensory inputs are defined by VA while bodily sensory inputs are defined by PA, which may result in a sensory neural mismatch.

The intent of ‘proxy studies’ is to evaluate the impact of PA on human health. To do so, these studies attempt to produce a virtual architectural setting that is as indistinguishable from reality as possible. The ‘lifelike’ visual nature of VA in proxy studies means that variations in gravitational perception are controlled for as best as possible. However, this will not always be the case, and indeed, an emerging style of VA (it should be acknowledged that, although ‘gravity defiant’ architectural structures are more likely to exist in VA due to the lack of gravitational restrictions, buildings that visually appear to be ‘gravity defiant’ do exist in PA as well. These buildings appear to be made possible by advances in construction technology [142] and result in designs that visually deceive the mind) intentionally defies ‘Earth gravity’. For example, Barcelona-based designer Andréas Reisinger has become an Instagram sensation, selling $450,000 USD worth of gravity-defying virtual structures and ‘impossible furniture’ [143]. However, the above research suggests that when a subject is exposed to visual cues of ‘earth discrepant gravity’ (i.e., a building which does not adhere to physical laws of Earth gravity), these conflicts between bodily and visual gravity cues inhibit the body’s ability to perceive Earth-discrepant gravities [144]. This suggests that exposure to VA may also result in neural mismatch and compromised gravitational perception, leading to physiological distress, and physiological dysfunction.

## 7. Summary of Issues

The preceding sections illustrate that, despite some limitations, VA may act as a suitable proxy to measure the health impacts of PA in architectural health research. These impacts may be positive, as illustrated by the positive benefits of virtual biophilic architecture established by Yeom et al. (2021) or negative, as illustrated by research on the negative stress responses to geometric room proportions established by Shemesh et al. (2021). However, this paper suggests that while findings from proxy studies may give some insight into the health implications of VA, they may not capture the full impact of VA on human health due to certain confounding variables unique to VA, particularly when exposure is for a prolonged period. This paper has identified two confounding variables that may further exacerbate the physiological impacts of VA: multisensory integration and gravitational perception. Figure 2 below illustrates the series of complex interactions between confounding variables in VA, proxy studies and PA health research.

## 8. Future Research

Future research, however, is needed in several areas. First, there is a pressing need for empirical validation of the hypothesis put forward in this paper [i.e., that the two identified confounding variables—multisensory integration and gravitational perception—may produce negative health impacts during long-term exposure to VA that are not captured by proxy studies]. To accomplish this, controlled studies could be conducted that isolate individual VA features (i.e., features related to multisensory integration and gravitational perception) and investigate their physiological implications using clinical biomarkers. Researchers have successfully captured the impacts of multisensory integration [145] and gravitational perception [146] using clinical biomarkers, such as single- and multi-unit responses, local field potentials, functional magnetic resonance imaging, electroencephalography, as well as behavioural measures of detection, accuracy, and response time. These methods may be transferred to the field of neuroarchitecture. Furthermore, it will be necessary for this research to take into account the impact of chronic or long-term exposure to these confounding features in VA. This will inevitably be complicated by methodological limitations related to participant exposure.

Second, while this paper has identified multisensory integration and gravitational perception as two potentially confounding variables, the scope of this study is limited, and it remains probable that additional variables exist. For example, the neurological activity involved in perceiving VR may differ. In an animal study by Aghajan et al. (2015), neurological activity in the brain area associated with spatial learning differed in virtual environments than in physical environments, with many neurons shutting down while in VR. The aetiology of these responses, however, remains unclear. This finding would suggest additional confounding variables pertinent to VA. Therefore, in addition to further empirical testing, a formal systematic review of the literature is recommended to gain a comprehensive understanding of all confounding variables unique to VA.

Third, it remains unclear how, if at all, these confounding variables interact with architectural variables examined in proxy studies e.g., biophilic design, lighting levels and room proportions. For example, how might the benefits of rectilinear and curvilinear architectural forms [106] combine with the potential physiological distress associated with compromised gravitational perception? Without studying VA in a way that takes these confounding variables into account, we cannot know the true impact of architectural forms on human health when experienced in virtual reality.

Fourth, future research may wish to consider whether or not the relative impact of the neurophysiological effects of VA on the purported sensory conflicts is meaningful when considering the net benefit of the stimuli presented in an experiment. Research on this is of particular interest given the emerging evidence examining the restorative uses of VA, such as those illustrated by Ashley Verzwyvelt (2021) in their study of oncology patients. With this in mind, it may be possible that the potential negative physiological consequences of VA can be mitigated by the introduction of ‘stress-reducing’ forms of VA.

Finally, future research may wish to examine how exposure to VA impacts psychosocial outcomes and behaviour. For example, does an increase in exposure to VA change the way in which individuals navigate PA? This line of enquiry may be of particular interest to those studying behavioural architecture and environmental psychology.

## 9. Conclusions

Emerging evidence suggests that we will see the rapid development and mainstream adoption of VA, facilitated by the incipient ‘metaverse’. It is therefore not only appropriate, but indeed critical, to consider public health concerns relating to extended exposure to VA. While studies directly measuring the impact of VA on human health have been limited, research conducted in the field of neuroarchitecture has increasingly used VA as a proxy by which to measure the impact of PA on human health. These ‘proxy studies’ have provided compelling evidence that acute exposure to PA may negatively or positively impact human health, depending on the circumstances. These studies’ findings may also give partial insight into the health impacts of VA. However, this paper questioned whether proxy studies are capable of fully capturing the health impacts of VA. An interdisciplinary survey of literature from the adjacent fields of sensory cognition and gravitational continuity identified multisensory integration and gravitational perception as two confounding variables which might limit the transferability of these proxy studies’ findings to VA. It was suggested that additional variables are also likely to exist, illustrating a promising area for future research. The presence of confounding variables suggests that neuroarchitectural health research has yet to fully capture the potential health concerns related to extended exposure to VA. Finally, further empirical testing is required to confirm the hypothesis put forward in this paper: that the two identified confounding variables—multisensory integration and gravitational perception—may produce negative health impacts during long-term exposure to VA that are not captured by proxy studies. In conclusion, the health concerns related to VA may be increasingly relevant for society, and the development of a new subfield within neuroarchitecture that empirically examines the potential health implications of extended exposure to VA is highly recommended.

## Figures and Tables

**Figure 1 ijerph-20-02735-f001:**
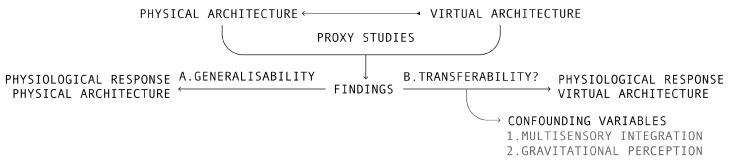
This proxy relational diagram illustrates the current state of the research examining the relational dynamics between proxy studies, physiological responses to PA and physiological responses to VA. Left: highlights the established generalisability between VA and PA, whereby VA is used to test physiological responses to PA. Right: illustrates the primary research question of this paper: can findings from proxy studies be transferred to understand physiological responses to VA? The diagram identifies the two proposed confounding variables, multisensory integration and gravitational perception, that problematise the prospect of transferability. Multisensory integration refers to the interaction between sensory inputs and the combination of their information content. While gravitational perception refers to the intersensory schema required to accurately perceive gravitational forces.

**Figure 2 ijerph-20-02735-f002:**
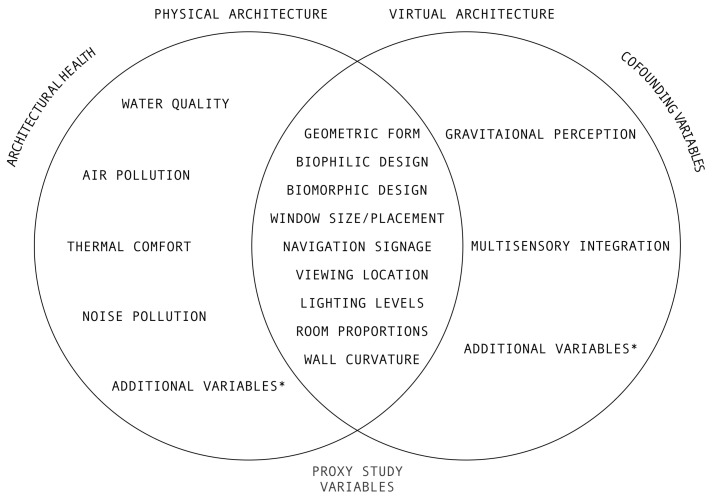
This Venn diagram illustrates the series of complex interactions between confounding variables in VA, proxy studies and PA health research. Left: variables (e.g., water quality and air pollution) that are directly relevant to occupant health and are inherent to PA. Centre: variables identified and assessed using proxy studies. These variables are relevant both to health concerns in PA (left) and in VA (as established by proxy studies). Right: confounding variables that may further exacerbate health concerns in VA (as identified in this paper). Both gravitational perception and multisensory integration are identified, however, additional variables are likely to exist. * Additional variables not considered by this manuscript.

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
