# Peer review of "Health Implications of Virtual Architecture: An Interdisciplinary Exploration of the Transferability of Findings from Neuroarchitecture"

_ijerph, 2023, doi:10.3390/ijerph20032735_

Round 1

Reviewer 1 Report

It's an interesting and well-written manuscript. I thought it should be a review rather than a research article. It includes several novel thoughts on virtual architecture and its health implications, but the methodology part is weak if it is a research article. Since the author identified two confounding variables: multisensory integration and gravitational perception, the manuscript needs to explain how these two variables are related to virtual architecture exposure and health outcomes.

I would also suggest the author propose some applications and directions of neuro-architectural health research.

Author Response

Dear Reviewer, 

Thank you for your thoughtful review of this manuscript. The current version of the document has been improved as a result of your insightful comments and valuable feedback. Please note that I have carefully considered each of your comments and have attempted to address them all. All modifications to the manuscript have been made using track changes in Overleaf. 

Please see a point-by-point response attached below.

Kind regards.

Reviewer 2 Report

1. The purpose of this study is clear, but the direction, definition, and nature of the research method are very ambiguous. It seems that the role of a very good tool in presenting the framework or basis of textbook theory at the level of general university textbooks can be well known, but the value of research as an original art is very insufficient.

2. The paper corresponding to the preparation of humanities systems and systems to implement VA should provide the basis for quantitative indicators derived to define these algorithms, and post-evaluation should be conducted with standardized evaluation tools with proven reliability and validity. In other words, in order to verify the VA derived by the authors, a tool or academic quantitative basis with solid validity used to produce it is urgently needed.

3. It is clear that research should not be limited and should be left to the researchers' autonomy, but it is common to include research produced based on solid theoretical and experimental research results, and inquiry methods to answer or verify research problems or hypotheses that include quantitative characteristics. Because it is a scientific technology.

4. This study is mainly an exploratory study and is a research method to interpret and understand the meaning of phenomena through the researcher's intuitive insight. It is mainly used to understand the underlying reasons, opinions, and motivations for a phenomenon. It is positive that it is expected to help develop ideas or hypotheses for quantitative research potentially.

Author Response

(The authors gave the same response as above.)

Reviewer 3 Report

A brief summary (one short paragraph) outlining the aim of the paper, its main contributions and strengths.

The intention of this paper is to consider the psychophysiological consequences of prolonged exposure to virtual architecture (VA), citing an increased societal need due to the emerging technologies that employ virtual reality, such as the metaverse. Furthermore, the paper argues the transferrability of findings from research that uses VA is problematic due to confounds such as  multisensory discontinuity and gravitational perception. Lastly, the author calls for the establihment of a subfield of neuroarchitectural health research that is focused on measuring the impacts of exposure itself. In my opinion, the greatest contribution of this paper is its suggestion that we consider the pathophysiological implications of using virtual reality to understand how humans experience virtual architecture which, as the author states, has much relevance to the cited potential transition to existing in a “metaverse” which necessitates a concerted scientific inquiry into the health impacts of prolonged exposure to virtual environments. The clear strength of this paper is its ambition in addressing a worthwhile issue involved in conducting neuroarchitectural (to use the author’s term) research that employs virtual reality. The author also does a good job of integrating diverse disciplines which highlights an importantly transdisciplinary approach to a complex problem. 

General concept comments 
Article: highlighting areas of weakness, the testability of the hypothesis, methodological inaccuracies, missing controls, etc.

-       A question to consider is whether the positive benefits of exposure to virtual environments makes up for the issues of multisensory integration and gravitational perception experienced. In other words, is the relative impact of the physiological effects of the purported sensory conflict meaningful when considering if the net benefit of the stimuli presented in an experiment.

Review: commenting on the completeness of the review topic covered, the relevance of the review topic, the gap in knowledge identified, the appropriateness of references, etc. 
These comments are focused on the scientific content of the manuscript and should be specific enough for the authors to be able to respond.

-       More detail and cited literature is required to establish the relevance of the identified confounds to virtual architecture.

-       In order for the general assertion of the article to be true it requires two assumptions:

o   That the confounds of gravitational perception and multisensory integration have a meaningful effect specifically on the experience of virtual architecture; this requires a more comprehensive backup from the literature given the tone and ambition of the article. I do not believe the sole paper cited to demonstrate the author’s argument.

·  Specific comments referring to line numbers, tables or figures that point out inaccuracies within the text or sentences that are unclear. These comments should also focus on the scientific content and not on spelling, formatting or English language problems, as these can be addressed at a later stage by our internal staff.

-       In the text box for Figure 1 it would be good to include definitions of the confounds so that the reader doesn’t need to search them out in the text to understand the figure

-       On line 57: “As this paper illustrates, humans may not exhibit the same physiological responses to the virtual world as to the physical world”. I would avoid assertions like this or simply indicate that “humans may or may not exhibit etc..”

-       On line 85 neurotechnology was referenced; it would be good to be more clear about the definition and examples.

-       Section 3 begins (line 121) with the following statement: “While failing to examine the direct impacts of prolonged exposure to VA on human health, the field of analytical neuroarchitecture has been heavily reliant on VA to evaluate the health impacts of visual exposure to PA.”

o   This is a very bold statement which requires a more coherent and comprehensive review of existing research, formally such as an actual literature review or a scoping review. It was not clear in the paper what method was used do the cited review on virtual architecture.

o   The term “analytical neuroarchitecture” was used to describe a field of research, however upon a google scholar (and google) search, nothing came up for the search term which is concerning given the field in question is being critiqued. I would suggest using existing terminology for the field you are criticizing, in order to have a constructive conversation about the issues you are highlighting.

-       Line 223: “Consequently, this paper argues that transferring the findings from proxy studies to the virtual world may prove problematic as proxy studies may be incapable of capturing the full physiological impact of VA due to confounding variables”. 

o   It is important to establish where in the research being critiqued where it is explicitly stated that proxy studies are exactly representative of the emualted real world environment. Environmental psychology, a necessary foundation for the identified field of “analytical neuroarchitecture: (AN) has done considerable research using two dimensional images to obtain insights on a three dimensional structure. Therefore, it would be good to provide an overview of ecological validity as it pertains to virtual architecture and the field of AN.

Author Response

(The authors gave the same response as above.)

Round 2

Reviewer 2 Report

The main focus has been stll pending, but if it is adequate for this field, the thesis may be able to this journal. 

Author Response

Dear Reviewer,

Thank you for your thoughtful review of this manuscript. This manuscript has been significantly improved as a result of your insightful comments and valuable feedback.

Kind regards. 

Reviewer 3 Report

Good job addressing the concerns identified. This paper will start an important and interesting conversation in the field of neuroarchitecture. Kudos!

Author Response

Dear Reviewer,

Thank you again for your thoughtful review of this manuscript. It is much appreciated! This manuscript has been significantly improved due to your insightful comments and valuable feedback.

Kind regards.